# Artificial intelligence–based technology for semi-automated segmentation of rectal cancer using high-resolution MRI

**Atsushi Hamabe**[1], **Masayuki Ishii**[1], **Rena Kamoda**[2], **Saeko Sasuga**[2], **Koichi Okuya**[1], **Kenji Okita**[1], **Emi Akizuki**[1], **Yu Sato**[1], **Ryo Miura**[1], **Koichi Onodera**[3], **Masamitsu Hatakenaka**[3], **Ichiro Takemasa**[1]*

1 Department of Surgery, Surgical Oncology and Science, Sapporo Medical University, Sapporo, Japan, 2 FUJIFILM Corporation, Tokyo, Japan, 3 Department of Diagnostic Radiology, Sapporo Medical University, Sapporo, Japan

* itakemasa@sapmed.ac.jp

**Data Availability Statement:** Raw data of MRI and pathological images contain potentially identifying patient information (patient-specific ID). Non-author contact information: the institutional review

## Abstract

### Aim

Although MRI has a substantial role in directing treatment decisions for locally advanced rectal cancer, precise interpretation of the findings is not necessarily available at every institution. In this study, we aimed to develop artificial intelligence-based software for the segmentation of rectal cancer that can be used for staging to optimize treatment strategy and for preoperative surgical simulation.

### Method

Images from a total of 201 patients who underwent preoperative MRI were analyzed for training data. The resected specimen was processed in a circular shape in 103 cases. Using these datasets, ground-truth labels were prepared by annotating MR images with ground-truth segmentation labels of tumor area based on pathologically confirmed lesions. In addition, the areas of rectum and mesorectum were also labeled. An automatic segmentation algorithm was developed using a U-net deep neural network.

### Results

The developed algorithm could estimate the area of the tumor, rectum, and mesorectum. The Dice similarity coefficients between manual and automatic segmentation were 0.727, 0.930, and 0.917 for tumor, rectum, and mesorectum, respectively. The T2/T3 diagnostic sensitivity, specificity, and overall accuracy were 0.773, 0.768, and 0.771, respectively.

### Conclusion

This algorithm can provide objective analysis of MR images at any institution, and aid risk stratification in rectal cancer and the tailoring of individual treatments. Moreover, it can be used for surgical simulations.

board of Sapporo Medical University Hospital.
E-mail: ji-rskk@sapmed.ac.jp.

**Funding:** This study was funded by the FUJIFILM Corporation (https://www.fujifilm.com). The funder had no role in study design, data collection and analysis, or decision to publish, but supported the analysis using deep learning and the preparation of manuscript related to deep learning. No authors received personal support from the funder.

**Competing interests:** The authors have declared that no competing interests exist.

## Introduction

In rectal cancer treatment, accurate diagnosis is crucial in determining individual treatment strategies and achieving curable resection. Multidisciplinary treatment including preoperative chemoradiotherapy is standard therapy for locally advanced rectal cancer (LARC) to prevent local recurrence after total mesorectal excision (TME), and here MRI has the pivotal role of defining the baseline stage of rectal cancer [1, 2]. ESMO and NCCN guidelines recommend MRI as a mandatory preoperative examination [3, 4].

Although the accuracy of MRI in predicting the stage of rectal cancer has been high in previous studies comparing MRI findings with histopathology in relatively small series, the MERCURY study that prospectively incorporated larger series did not replicate the prior excellent results [5–11]. In addition, when expert radiologists interpreted the MR images according to strictly defined protocols, satisfactory accuracy was maintained, but this is not necessarily the practice at every institution [12]. Other possible concerns include inter-observer differences in difficult cases, or the shortage of specialized radiologists in some developed countries [13, 14]. If a system supporting MRI diagnosis could be implemented, it would be useful in many circumstances.

Recent progress in applied artificial intelligence (AI) has increased its importance in medical care, especially in medical image analysis [15–17]. The use of AI-based diagnostic supporting technology is enabled by advances in deep learning technology (DL). With the use of a substantial number of high-quality training datasets, DL can make an algorithm that predicts clinical output with high accuracy. Ronneberger et al. introduced the U-net for the segmentation of two-dimensional (2D) biomedical images [18], and Milletrai et al. extended the U-net to three-dimensional (3D) images [19]. Regarding tumor segmentation from MR images, the previous studies used these 2D or 3D U-nets and showed that the results of segmentation were comparable to those achieved by human experts in multiple types of cancer [20, 21]. While there have been several studies attempting to segment rectal cancers, the depth of tumor invasion could not be assessed or the accuracy of segmentation could stand further improvement [22, 23]. We have performed the PRODUCT study (UMIN000034364), in which we measured the circumferential resection margin (CRM) of LARC as a primary endpoint in laparoscopic surgery. Resected specimens including rectal cancer were processed in a circular shape with mesorectum attached for pathological diagnosis, though this has not been the general practice in Japan. In addition, we started to measure CRM according to the practice in Western countries, not only in the cases enrolled in the PRODUCT study but also in other LARC cases as a clinical practice. As a spin-off, available sections of these specimens show the areas of LARC that correspond to the MR images, thus providing high-quality training datasets which we consider advantageous in making ground-truth labels that can be used for DL.

Based on this background, we hypothesized that DL might resolve the difficulties related to MRI diagnosis by using MR images annotated with ground-truth labels reflecting the pathologically proved cancer area. In this study, we aimed to develop AI-based software to support the staging diagnosis of rectal cancer and to visualize the segmentation of rectal cancer, which can be used to optimize treatment strategy and in surgical simulations.

## Materials and methods

### Patients

The patients who underwent surgery for rectal cancer between January 2016 and July 2020 in our institution were retrospectively analyzed (Fig 1). A total of 201 MRI exams were used for training data (Table 1). Of these, a resected specimen was processed in a circular shape in 103

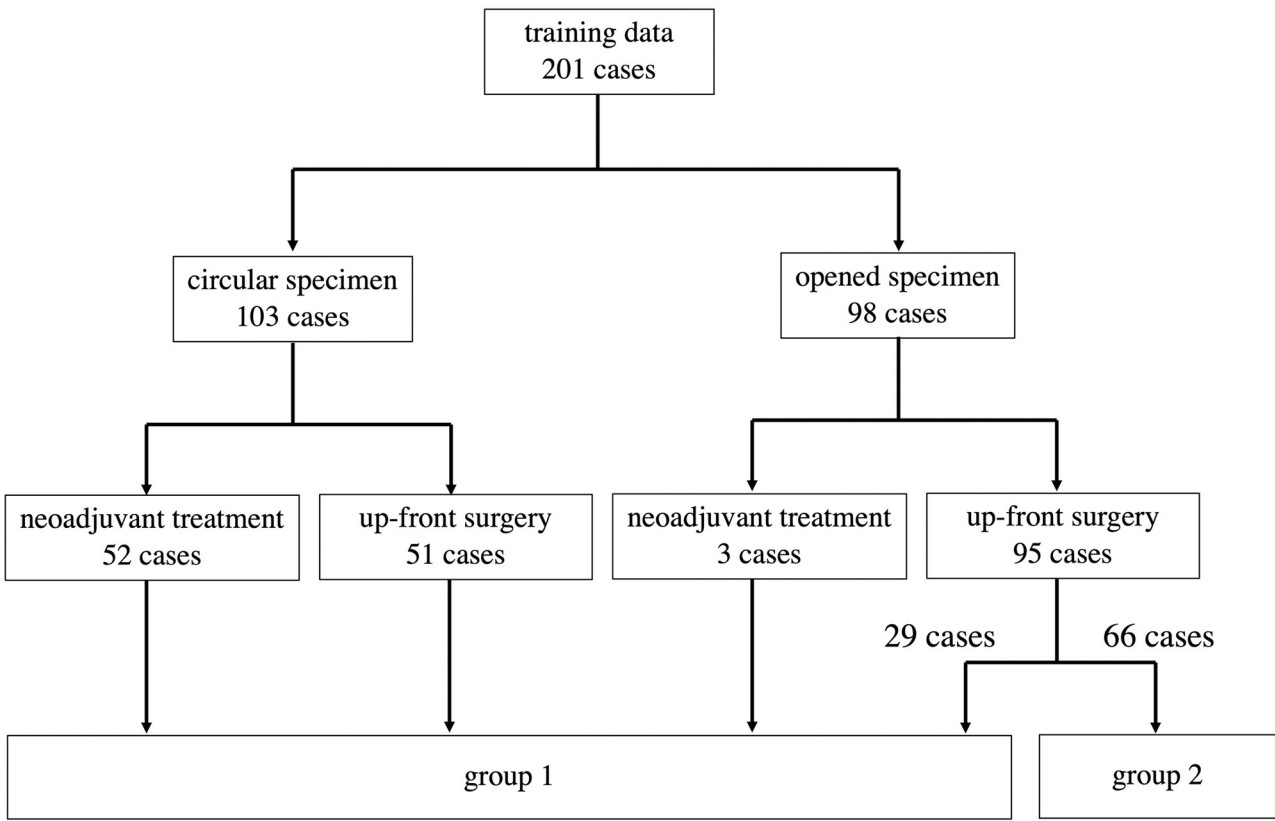

**Fig 1. Details of a total of 201 cases used as training data.** Group 1 images were used to prepare ground-truth labels for segmentation. Group 2 images were used as ground-truth labels having pathological information of T staging alone.

cases, and neoadjuvant treatment was administered in 55 cases. A total of 98 opened specimens in which mesorectum was detached according to the standard Japanese procedure were included in the analysis. The protocol for this research project was approved by the Ethics Committee of Sapporo Medical University. Informed consent was not required due to the fact that data was anonymized. The procedures were in accordance with the provisions of the Declaration of Helsinki of 1995 (as revised in Brazil, 2013).

## Magnetic resonance imaging

MR images were acquired using a 3.0-T (N = 93) or 1.5-T (N = 108) MR scanner (Ingenia; Philips Healthcare, Best, the Netherlands). A phased-array coil (dStream Torso coil; Philips Healthcare, Best, the Netherlands) was used for signal reception. In 4 patients who were referred from the other hospitals, different MR scanners were used (3.0-T Skyra; Siemens,

**Table 1. Summary of the analyzed cases.**

|  | N = 201 |
|---|---|
| Sex (male/female) | 115/86 |
| T factor ($\leq$T2/T3/T4) | 82/103/16 |
| Neoadjuvant treatment (yes/no) | 55/146 |
| Processing for pathological examination (circular/open) | 103/98 |

Erlangen, Germany in 2 and 1.5-T Signa HDXt; GE Healthcare, Cleveland, OH, USA in 2, respectively). Before examination, bowel peristalsis was prevented by intramuscular injection of butylscopolamine if possible. Neither bowel preparation nor air insufflation was performed. After identifying the tumor on sagittal T2-weighted images, axial T2-weighted images were acquired in which the angle of the plane was made perpendicular to the long axis of the tumor (TR/TE, 4000/90 ms; 3-mm slice thickness; 0.5-mm interslice gap; 150-mm field of view; $288 \times 288$ matrix; spatial resolution, $0.52 \times 0.52$ pixel size). Three-dimensional isotropic T2-weighted fast spin-echo was also acquired routinely since October 2018 (TR/TE, 1500/200 ms; 256-mm field of view; $288 \times 288$ matrix; spatial resolution $0.89 \times 0.89$ mm).

## Processing of resected specimen

In the PRODUCT study, we developed a new method to precisely measure the pathological CRM, which we named "transverse slicing of a semi-opened rectal specimen" [24]. First, the anterior side of the rectum is opened longitudinally from the oral stump to the anal side up to 2 cm oral to the tumor border. Similarly, the rectum is opened on the anal side to the tumor if sufficient distal margin is resected. That is, the area of rectum between 2 cm above and below the borders of the rectal cancer is not incised. The mesorectum attached to the opened region of the rectum is removed to harvest embedded lymph nodes, while the mesorectum is left attached where the rectum is not opened. After the removal of the mesorectum, the dissection plane is marked using India ink for the purpose of demarcating it and supporting CRM measurement. Next to the inking, a piece of soft sponge is inserted in the rectal lumen to keep the in situ circular shape and the specimen is pinned to a cork board under gentle tension, followed by fixation in 10% formalin. After fixation, a circular area of the rectum is transversely sliced as thinly as possible. Pathologists analyzed all sections after staining with hematoxylin-eosin and diagnosed pathological findings.

## Ground-truth label

Since we use a supervised training method to develop automatic segmentation algorithms, ground-truth labels were required. For all 201 cases, baseline T stages were labeled based on the pathological diagnosis or on the assessment of pathological sections if the patients had undergone neoadjuvant treatment. Segmentation labels, which represent whether each voxel of an MR image belongs to the target subject or not, were prepared for 135 of the 201 cases by two surgeons (AH and MI) who each has more than 10 years' clinical experience treating colorectal cancer. Before starting the analysis, they received several lectures from a qualified pathologist to train them to find the area of rectal cancer or to predict the baseline area of rectal cancer before neoadjuvant treatment by discriminating fibrosis or necrosis on hematoxylin and eosin sections. These surgeons created MR images annotated with ground-truth segmentation labels, including the areas of tumor, rectum, and mesorectum, using 3D MRI analysis software (Fig 2). The rectal area was defined as the area within the muscularis propria.

## Automatic segmentation algorithm

We developed an automatic segmentation algorithm that extracts the tumor, rectum, and mesorectum areas in 3D from T2-weighted MR images using a deep neural network. The network architecture is a 3D variant of U-net, which is popular for biomedical image segmentation [18]. It consists of encoder and decoder parts with skip connections (Fig 3). The convolutional block in each encoder and decoder consists of a $3 \times 3 \times 3$ or $1 \times 3 \times 3$ convolution layer, a batch normalization layer, and rectified linear unit operations. The deconvolution blocks are transposed convolutional operators with a kernel size of $4 \times 4 \times 4$ voxels. The skip

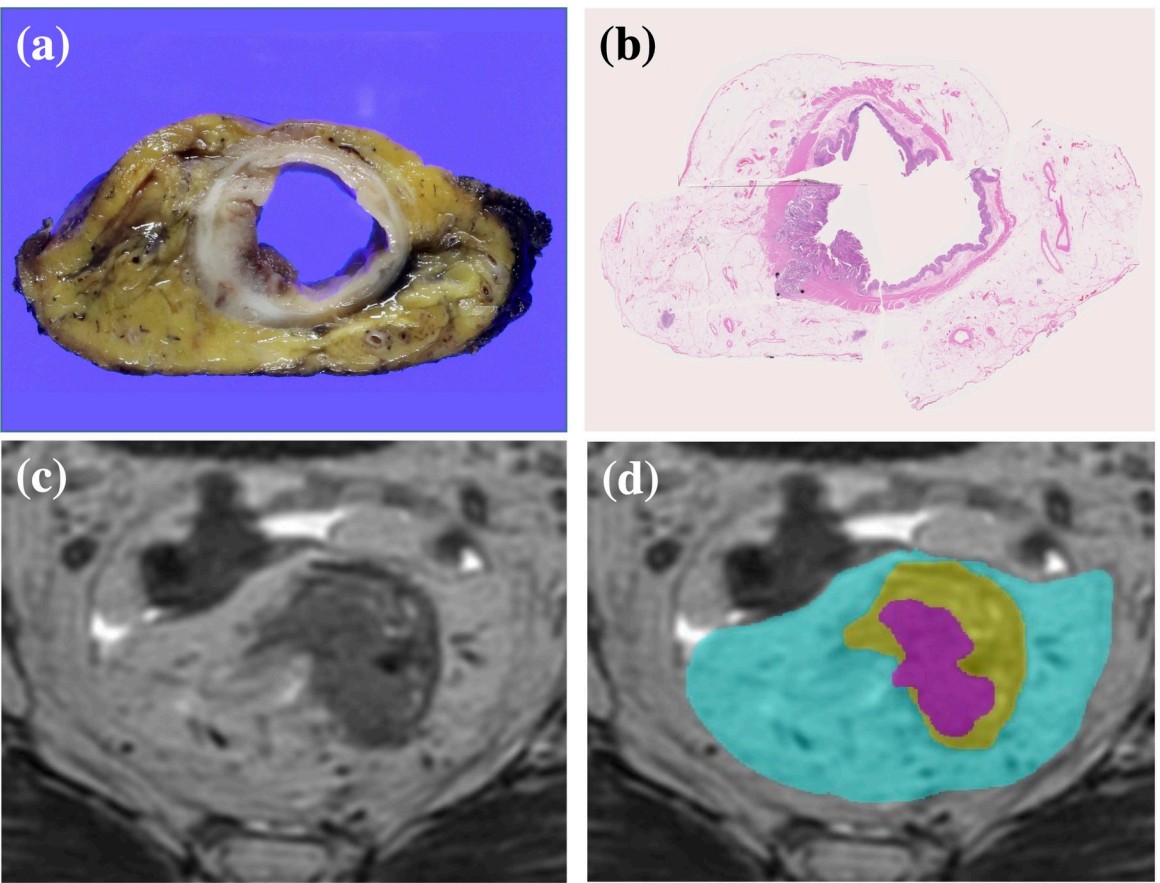

**Fig 2. Preparation for ground-truth segmentation labels.** (a) Section of a circular specimen. (b) Pathological section of the specimen stained with hematoxylin-eosin revealing areas of tumor, rectum, and mesorectum. (c) Axial MR image of the rectal cancer. (d) Ground-truth segmentation labels were used to annotate the MR images. The areas colored magenta, yellow, and cyan represent tumor, rectum, and mesorectum, respectively.

connections include a $1 \times 1 \times 1$ convolution layer, a batch normalization layer, and rectified linear unit operations. The input to the network is a 3D MR image. The output has same spatial dimensions as the input, with 3 channels each for the mesorectum area, rectum, and tumor area probabilities. The last three channels have values from 0.0 to 1.0 with application of the sigmoid function. Final segmentation results were obtained by binarizing the values, using a threshold of 0.5.

Our algorithm calculates the T stage, following the binary segmentation results. The case is classified as T2 or below when the tumor area is not in contact with the contour of the area of the rectum and completely included in the area of the rectum. Otherwise, the case is classified as T3 or above when at least a part of the tumor area is outside the rectum. This rule exactly follows the T staging rules of tumor invasion into the area of the rectum (Fig 4). Generally, the DL-based segmentation method works to maximize the volume overlap between the segmentation result and the ground-truth label image. However, the risk of disagreement for T-staging would be inherent if T-staging were based on segmentation results of tumor and rectum that were mutually independent. To deal with this concern, we introduced a novel loss that can directly maximize T-staging accuracies in model training. The loss consists of two terms, as follows. The first term is so-called Dice loss [19], which for segmentation purposes is

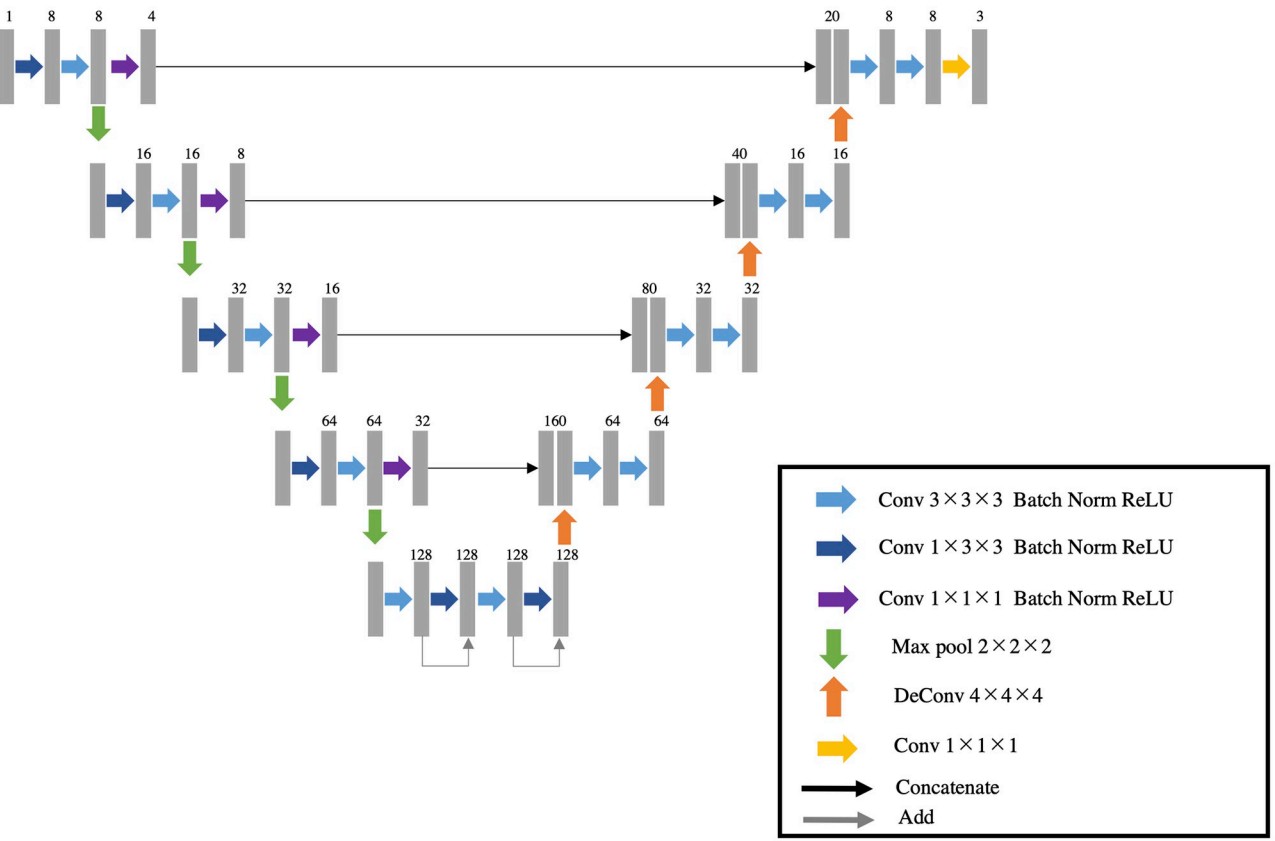

**Fig 3. U-net.** The architecture of the segmentation network for the areas of tumor, rectum, and mesorectum.

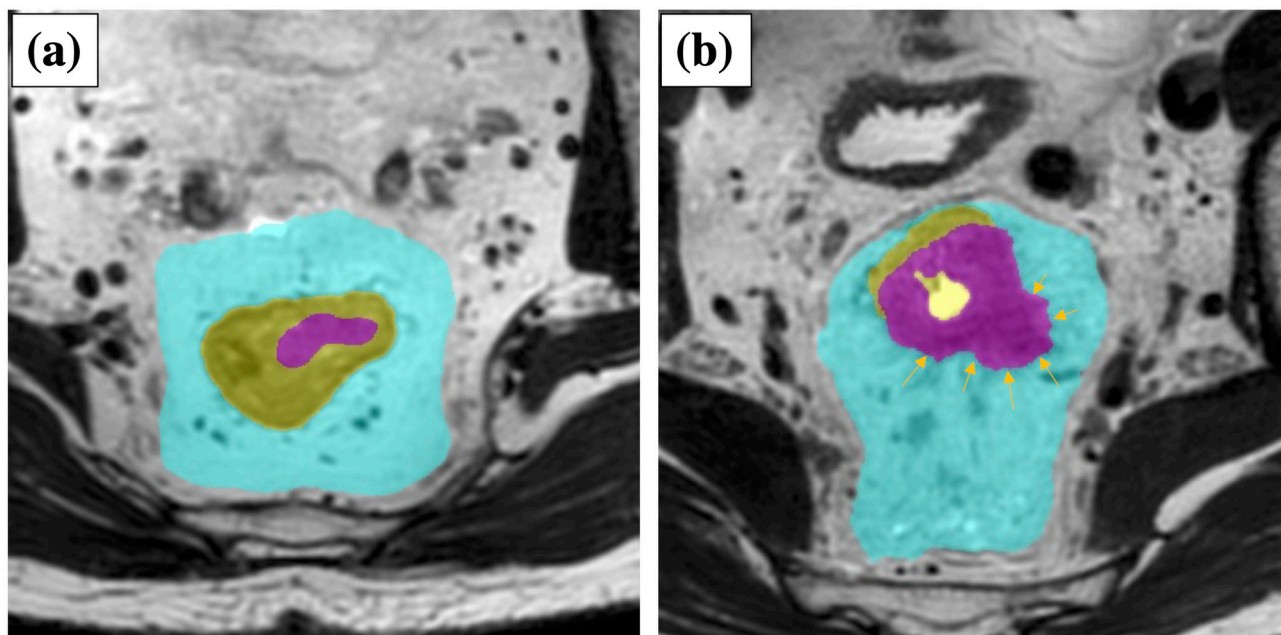

**Fig 4. Staging algorithm.** Left, T2 case, and right, T3 case. The magenta, yellow, and cyan areas represent tumor, rectum and mesorectum, respectively.

defined as follows:

$$Loss_{SEG} = 1 - \frac{2 \times \sum_{i=1}^{N} p_i g_i}{\sum_{i=1}^{N} p_i + \sum_{i=1}^{N} g_i}$$

where $N$ is the number of voxels, $p$ is the probability that is outputted by the network, and $g$ is the ground-truth label. This term works to maximize the overlap between the ground-truth label and the probability maps.

The second term of the loss function is cross entropy loss, which for accurate staging purposes is defined as follows:

$$Loss_{STG} = -\left(\left(1 - g_{staging}\right)/2 + g_{staging} \times p_{staging}\right)$$

where

$$p_{staging} = \max_{i \in \{1,...,N\}} \left(p_{cancer,\ i} \times \left(1 - p_{rectal\ tube,\ i}\right)\right),$$

$$g_{staging} = \begin{cases} 1 & \text{if ground truth T stage is over T3,} \\ -1 & \text{otherwise.} \end{cases}$$

$p_{cancer}$ and $p_{rectal\ tube}$ represent the probability maps of the tumor and rectum, respectively. $p_{staging}$ indicates the probability of the predicted staging. It takes a high number when there is any voxel simultaneously having low rectum probability and high tumor probability. This term works to reduce the tumor area outside of the rectum for T2 cases. On the other hand, it works to increase the tumor area outside of the rectum for T3 cases.

To summarize, we minimize the loss function to train the network:

$$Loss_{SEG} + \lambda \times Loss_{STG}$$

$\lambda$ is a parameter used to balance the two terms and it was experimentally determined to be 0.02. During the training, $Loss_{SEG}$ is evaluated only for the cases with ground-truth segmentation labels, while $Loss_{STG}$ is evaluated for all cases. We used the Adam optimizer to minimize the loss function, with the following parameters: base learning rate, 0.003; beta1, 0.9; beta2, 0.999; and epsilon, $1 \times 10^{-8}$. The batch size was 5 samples, including 3 cases with ground-truth segmentation labels and 2 cases with only ground-truth staging. All experiments were conducted on an NVIDIA DGX-2 machine using the NVIDIA V100 GPU with 80 GB of memory.

In the network training, each training image is augmented by several image-processing techniques such as scaling, rotation, and slice thickness conversion to improve segmentation accuracies. Also, the input image is cropped around the tumor area and rescaled to a 0.5 mm$^3$ isotropic voxel size and $256 \times 256 \times 128$ voxel number. In the test phase, a user inputs an estimated center position of the tumor, and then the image around the tumor position is processed.

## Workflow for evaluation and statistical analysis

We evaluated two aspects of the algorithm: segmentation accuracy and staging accuracy. Ten-fold cross validation was conducted. The data were randomly divided into 10 datasets. Eight datasets out of 10 were used for training the network parameters. The remaining two datasets were used for validation and evaluation, respectively. During the training iteration, the performance of the network was evaluated at every 100th iteration on the validation dataset. We chose the best network parameter for the validation dataset, using the sum of the dice score,

sensitivity, and specificity, and then applied it to the evaluation dataset. We repeated this procedure ten times, changing the role of training, validation, and evaluation of each dataset.

Regarding the segmentation accuracy, we calculated the Dice similarity coefficients (DSC) between manual segmentation and automatic segmentation [25]. The DSC is defined as follows:

$$DSC = \frac{2 \times |P \cap G|}{|P| + |G|}$$

where $P$ is the segmentation result and $G$ is the ground truth. The DSC ranges from 0.0 to 1.0, and DSC = 1.0 means that the results overlap completely. Note that, since not all of the training data have corresponding ground-truth segmentation, we evaluated the segmentation accuracies using 135 cases.

Next, the T staging accuracies were evaluated with all 201 cases by calculating the sensitivity and specificity. The sensitivity is defined as follows:

$$Sensitivity = \frac{|P_{T3} \cap G_{T3}|}{|G_{T3}|}$$

where $P_{T3}$ represents the predicted T stage as being over T3. $G_{T3}$ represents the ground-truth T stage as being over T3. Specificity is defined as follows:

$$Specificity = \frac{|P_{T2} \cap G_{T2}|}{|G_{T2}|}$$

where $P_{T2}$ means the predicted T stage is under T2 and $G_{T2}$ is means the ground-truth T stage is under T2.

Results are presented as the number of cases evaluated for categorical data and expressed as the median and interquartile range (IQR) for quantitative data. Univariate analysis was performed using the Wilcoxon rank-sum test. Statistical analyses were performed using JMP Pro 15.1.0 software (SAS Institute, Cary, NC, USA).

## Results

### Segmentation accuracy

The developed algorithm could successfully estimate the areas of the tumor, rectum, and mesorectum, in which the ground-truth labels and segmentation results of typical cases corresponded well (Fig 5a). The summary of evaluation results regarding the segmentation accuracy demonstrated that the median DSCs for tumor, rectum, and mesorectum were 0.727, 0.930, and 0.917, respectively (Fig 5b). Mucinous cancer exhibits high intensity on T2 in contrast to the most common histology of adenocarcinoma. Therefore, we investigated DSCs in mucinous cancer patients (N = 6) to analyze whether this feature affects segmentation accuracy. As a result, the DSC was lower in the cases of mucinous cancer compared with those of the other histology (0.358 [0.167–0.596] vs 0.736 [0.605–0.801], P = 0.0024). In addition, on the assumption that the DSC of the tumor might easily have been lowered by a slight positional deviation in the smaller tumor, the correlation between the DSC and the diameter of the tumor was investigated after excluding mucinous cancer (Fig 5c). We then observed a significant correlation between the two values (Pearson correlation coefficient = 0.2418; P = 0.0081). After excluding cancers of diameters less than 20 mm, the median DSC of the tumor was slightly elevated, to 0.739 [0.615–0.801].

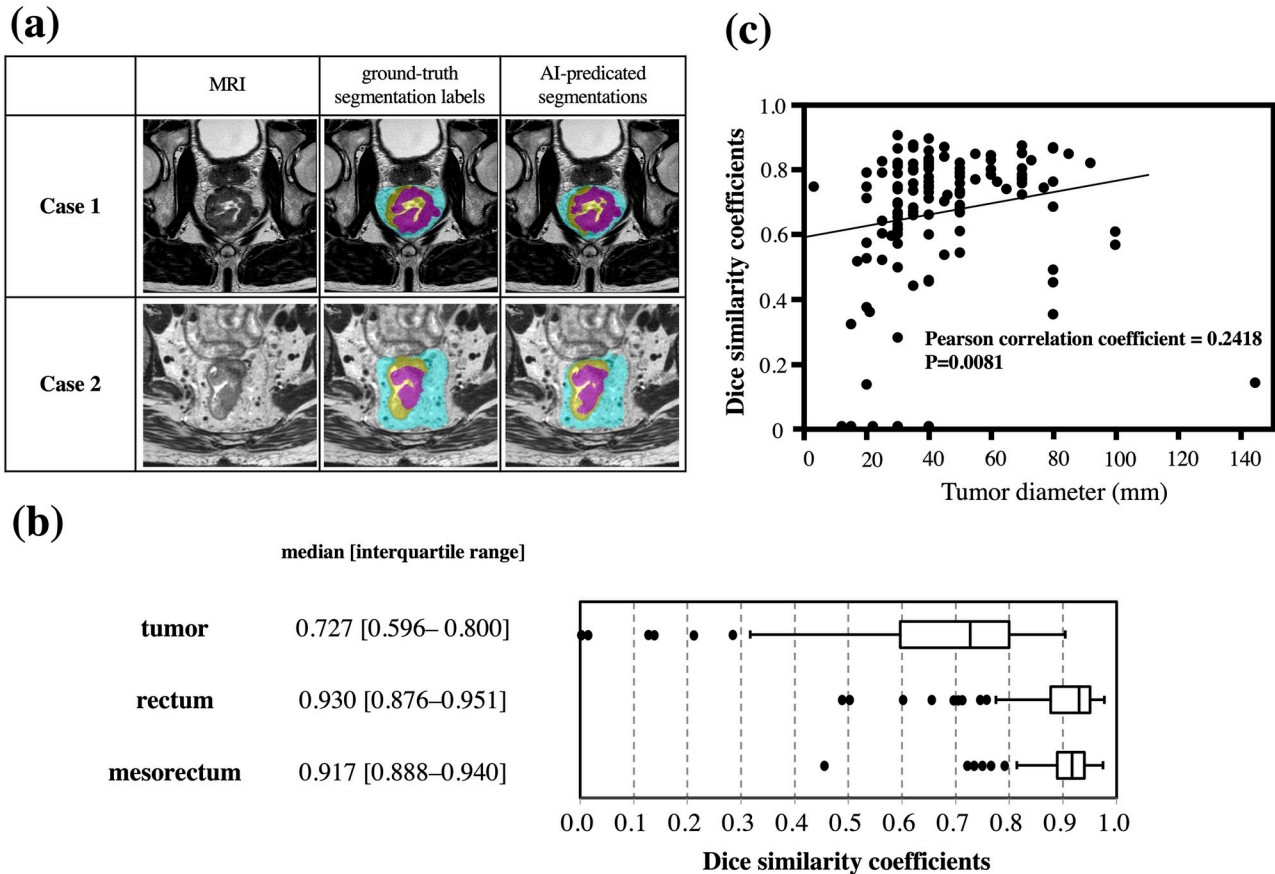

**Fig 5. Results of segmentation accuracy.** (a) Representative images of MRI, the ground-truth segmentation labels, and AI-predicated segmentations. (b) Summary of evaluation results regarding the segmentation accuracy. (c) Scatter plots showing the relationships between tumor diameter and the Dice similarity coefficients.

## Correlation between pathological and AI T stage

The guidelines used worldwide regard distinguishing between T2 and T3 as one of the important factors directing treatment decisions. Therefore, we investigated our method's diagnostic accuracy in discriminating T2 from T3 as an initial assessment. The summary of correlation between pathological T stage and AI-predicted T stage was analyzed (Table 2). The T-staging sensitivity, specificity, and overall accuracy were 0.773, 0.768, and 0.771, respectively. For comparison, we evaluated a baseline model that was trained by using a standard dice loss with only ground-truth segmentation labels. The baseline model obtained a sensitivity, specificity, and overall accuracy of 0.765, 0.756, and 0.761, showing that the AI developed in this study could

**Table 2. Summary of pathological T stage and AI-predicted T stage.**

| | | Ground-truth pathological T staging | | |
|---|---|---|---|---|
| | | ≤T2 | ≥T3 | Total |
| AI-predicted T staging | ≤T2 | 63 | 27 | 90 |
| | ≥T3 | 19 | 92 | 111 |
| Total | | 82 | 119 | 201 |

achieve better performance in T-staging. As in the analysis of segmentation accuracy, the diagnostic accuracy was recalculated after the exclusion of small cancers and mucinous cancer. As a result, the T-staging sensitivity, specificity, and overall accuracy were 0.789, 0.714, and 0.762, respectively.

## Discussion

In this study, an algorithm for diagnosing and staging rectal cancer was successfully developed using DL technology. It could be used in future semi-automation software to aid physicians. The characteristic feature of this algorithm is that it can output the segmentation that visualizes the areas of tumor, rectum, and mesorectum. This could be used not only for T-factor staging, but also for preoperative surgical simulation. In the future, based on the provided visual information, we will be able to choose the surgical plane to be dissected or decide whether the combined resection of an adjacent organ is necessary. In addition, we think the algorithm will also help multidisciplinary teams tailor treatment to individual patients.

Two meta-analyses have investigated the diagnostic accuracy of MRI and shown favorable results, with about 85% sensitivity and 75% specificity for diagnosing tumor invasion beyond the muscularis propria [10, 11]. However, these results are subject to substantial selection bias, which can be associated with higher reported than actual accuracy. This is partly reflected by the fact that the carefully designed prospective study, MERCURY, demonstrated diagnostic accuracy that was acceptable but that did not reach the values reported in the meta-analyses. Accurately diagnosing rectal cancer using MRI would, in reality, not be easy. Furthermore, although MRI scanners are plentiful in Japan, certified radiologists are in quite short supply, leaving individual radiologists with excessive workloads. This is also the case in other developed countries [13, 14]. Given this situation, a method that can improve the acquisition of objective MRI findings at every institution is needed. We think the current algorithm might play a substantial role in providing equal access to MRI diagnosis in institutions or regions where there are shortages of trained personnel.

As MRI technology has advanced in recent decades, it is important to re-evaluate the accuracy of MRI. Since neoadjuvant CRT was established as a standard treatment in Western countries, it has become difficult to validate the accuracy of baseline MRI findings by simply comparing them with the corresponding pathology. In the current study, we made a training dataset by annotating the pathologically proven tumor areas on MRI images. In the cases with neoadjuvant therapy, the baseline area of the tumor was predicted by the pathological evidence of fibrosis or necrosis. These processes might be useful in making reliable training datasets even in cases with neoadjuvant treatment, suggesting that the algorithm for segmentation might reflect the typical results of MRI today.

Some recent studies have tried to estimate rectal cancer–related parameters on preoperative MR images using AI, and have shown that the accuracy was acceptable [22, 26–28]. However, these studies had several limitations: tumor tissue was not visualized on the MR image, the relationship of the tumor with the mesorectal fascia was difficult to assess, the results were not based on high-resolution MRI, or the ground-truth labels were not based on pathological assessment, the last issue being the one we consider to be most critical. We think there is much room for improvement in the clinical application of AI. However, the software developed in this study has various strengths. First, the ground-truth labels are based on the pathological findings in circular specimens, providing the high-quality training datasets that are essential in establishing a reliable algorithm. Second, the algorithm can output the segmentation of the tumor, rectum, and mesorectum. This feature is valuable for staging the tumor, for individual multidisciplinary treatment decision making, and for the preoperative simulation that is

required by colorectal surgeons in order to obtain curative resection. Third, we used high-resolution MRI in this analysis, though the MRI acquisition protocols differ from those used in the MERCURY study. Thus, this system can be applied anywhere if the appropriate protocol and an adequate scanner are used for image acquisition. We note that the accuracy of our algorithm was insufficient in analyzing some types of tumors, including mucinous cancer and small tumors. Although the quality of segmentation can also be regarded as favorable as a whole, it would be ideal if these hurdles were cleared with future refinement. However, because these small tumors rarely infiltrate the mesorectum or surrounding tissues, this algorithm can still be regarded as useful for diagnosing locally advanced rectal cancers.

The current study has several limitations. First, validation using the test data acquired in various conditions should be performed to confirm the generalizability of the algorithm. Currently, we are planning a validation study using an independent large series to investigate the algorithm's effectiveness. Simultaneously, we will continue to improve the software's performance in assessing other important factors, including mesorectal fascia involvement. Second, the workload involved in preparing individual ground-truth labels is too heavy for the number of training sets to be readily increased. Third, as explained in the results, the accuracy of this system is still insufficient to be used for mucinous tumors and it is not able to estimate the shape of small tumors. We think this limitation can be overcome with the use of more training datasets in the future.

In conclusion, we have successfully developed the first AI-based algorithm for segmenting rectal cancer. This system can provide stable results at any institution and contribute to rectal cancer risk stratification and the tailoring of individual treatments, and is likely to gain importance in the era of individualized medical care.

## Supporting information

**S1 Dataset.**
(XLSX)

## Acknowledgments

We are grateful to Shintaro Sugita, Associate Professor in the Department of Surgical Pathology at Sapporo Medical University, for giving lectures on finding areas of rectal cancer prior to preparing ground-truth labels.

## Author Contributions

**Conceptualization:** Atsushi Hamabe, Masayuki Ishii, Koichi Okuya, Masamitsu Hatakenaka, Ichiro Takemasa.

**Formal analysis:** Atsushi Hamabe, Masayuki Ishii, Koichi Okuya, Kenji Okita, Emi Akizuki, Yu Sato, Ryo Miura, Koichi Onodera.

**Funding acquisition:** Atsushi Hamabe, Ichiro Takemasa.

**Investigation:** Atsushi Hamabe, Masayuki Ishii, Kenji Okita, Emi Akizuki, Yu Sato, Ryo Miura, Koichi Onodera.

**Methodology:** Atsushi Hamabe, Masayuki Ishii, Rena Kamoda, Saeko Sasuga, Koichi Okuya, Kenji Okita, Koichi Onodera, Masamitsu Hatakenaka, Ichiro Takemasa.

**Resources:** Atsushi Hamabe, Masayuki Ishii, Emi Akizuki, Yu Sato, Ryo Miura, Koichi Onodera.

**Software:** Rena Kamoda, Saeko Sasuga.

**Supervision:** Masamitsu Hatakenaka, Ichiro Takemasa.

**Writing – original draft:** Atsushi Hamabe, Masayuki Ishii, Rena Kamoda, Saeko Sasuga, Koichi Okuya, Koichi Onodera.

**Writing – review & editing:** Kenji Okita, Masamitsu Hatakenaka, Ichiro Takemasa.

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
