## [Decision Letter · Decision Letter 0]

2 Feb 2022

PONE-D-21-25683Artificial intelligence–based technology for semi-automated segmentation of rectal cancer using high-resolution MRIPLOS ONE

Dear Dr. Takemasa,

Thank you for submitting your manuscript to PLOS ONE. After careful consideration, we feel that it has merit but does not fully meet PLOS ONE’s publication criteria as it currently stands. Therefore, we invite you to submit a revised version of the manuscript that addresses the points raised during the review process.

We look forward to receiving your revised manuscript.

Kind regards,

Kumaradevan Punithakumar

Academic Editor

PLOS ONE

https://journals.plos.org/plosone/s/file?id=ba62/PLOSOne_formatting_sample_title_authors_affiliations.pdf"

3. Please clarify the nature of these restrictions, ie. If due to ethical or legal reasons.

4. Please amend your Methods section to include the information provided in your Ethics Statement that informed consent was not required due to the fact that data was anonymized.

5. Thank you for stating the following in the Funding Section of your manuscript:

“This study was funded by the FUJIFILM Corporation”

“This study was funded by the FUJIFILM Corporation (https://www.fujifilm.com). The funder had no role in study design, data collection and analysis, or decision to publish, but supported the analysis using deep learning and the preparation of manuscript related to deep learning. No authors received personal support from the funder.”

Reviewers' comments:

Reviewer's Responses to Questions

**Comments to the Author**

1. Is the manuscript technically sound, and do the data support the conclusions?

Reviewer #1: Yes

Reviewer #2: Yes

2. Has the statistical analysis been performed appropriately and rigorously? 

Reviewer #1: Yes

Reviewer #2: Yes

3. Have the authors made all data underlying the findings in their manuscript fully available?

Reviewer #1: No

Reviewer #2: Yes

4. Is the manuscript presented in an intelligible fashion and written in standard English?

Reviewer #1: Yes

Reviewer #2: Yes

5. Review Comments to the Author

Reviewer #1: The entitled "Artificial intelligence–based technology for semi-automated segmentation of rectal

cancer using high-resolution MRI " is well organized. Literature review and methodology explained by author is appreciable. minor revision is suggested. Author should address following queries:

1.in introduction section(3rd paragraph), some recent papers of this area should be covered which uses U-net in MRI images or other techniques apart from your used technique.

2.In method section, automatic segmentation algorithm part, Author should add more detail about the algorithm and qualitatively analyse the scales of images before and after the application of segmentation algorithm.

3.In line 148, proper citations are missing. Please place the appropriate citations for evaluation coefficients.

4. in figure 5, pearson correlation coefficient and p value should be noted in the figure,and the y-axis is fuzzy.

Reviewer #2: This manuscript presents a new high-resolution MRI semi-automatic rectal cancer segmentation technology based on deep learning. I believe this work will contribute to the accuracy of adjuvant therapy. However, I hope the author will revise the manuscript and solve the following problems before publishing it in the journal.

1. Firstly, the description of the research status is not perfect, especially the research status of deep learning in medical image aided diagnosis, including two-dimensional image and three-dimensional image.

2. Secondly, it is necessary to make a detailed description of the hardware configuration and super parameter configuration of network training.

3. Finally, some other networks should be added to the experimental part as a contrast, so as to make the experimental results more convincing.

6. PLOS authors have the option to publish the peer review history of their article (what does this mean?). If published, this will include your full peer review and any attached files.

Reviewer #1: No

Reviewer #2: No

---

## [Author Response · Author response to Decision Letter 0]

11 Apr 2022

Responses to the reviewers’ comments

We thank the reviewers for their fair comments and useful suggestions for improving our manuscript. As indicated below, we have considered all comments and suggestions, and we made corrections on a proof in red for the revised parts of the manuscript.

Reviewer's Responses to Questions

Comments to the Author

3. Have the authors made all data underlying the findings in their manuscript fully available?

Reviewer #1: No

Reviewer #2: Yes

Response: According to the suggestion, we uploaded the underlying dataset for the findings as supporting information files.

Reviewer #1: 

The entitled "Artificial intelligence–based technology for semi-automated segmentation of rectal cancer using high-resolution MRI " is well organized. Literature review and methodology explained by author is appreciable. minor revision is suggested. Author should address following queries:

1.In introduction section (3rd paragraph), some recent papers of this area should be covered which uses U-net in MRI images or other techniques apart from your used technique.

Response: 

We thank the reviewer for the constructive comment. We updated the current status regarding the U-net based segmentation techniques for MRI in introduction section (page 7, line 99-105). We additionally cited the following 6 references herein.

#18. Ronneberger O, Fischer, P., Brox, T. U-Net: Convolutional Networks for Biomedical Image Segmentation. arXiv:1505.04597 2015.

#19. Milletari F, Navab, N., Ahmadi, S.A. V-Net: Fully Convolutional Neural Networks for Volumetric Medical Image Segmentation. arXiv:1606.04797.

#20. Dolz J, Xu X, Rony J et al. Multiregion segmentation of bladder cancer structures in MRI with progressive dilated convolutional networks. Med Phys 2018; 45: 5482-5493.

#21. Hodneland E, Dybvik JA, Wagner-Larsen KS et al. Automated segmentation of endometrial cancer on MR images using deep learning. Sci Rep 2021; 11: 179.

#22. Trebeschi S, van Griethuysen JJM, Lambregts DMJ et al. Author Correction: Deep Learning for Fully-Automated Localization and Segmentation of Rectal Cancer on Multiparametric MR. Sci Rep 2018; 8: 2589.

#23. Huang YJ, Dou Q, Wang ZX et al. 3-D RoI-Aware U-Net for Accurate and Efficient Colorectal Tumor Segmentation. IEEE Trans Cybern 2021; 51: 5397-5408.

2. In method section, automatic segmentation algorithm part, Author should add more detail about the algorithm and qualitatively analyse the scales of images before and after the application of segmentation algorithm.

Response:

According to the indication, we added proposed algorithm features and implementation details (page 14, line 213-218). The salient feature of the algorithm is that we introduced a novel loss which can directly maximize T-staging accuracies in model training, thereby showing better performance than usual segmentation methods which try to optimize the volume overlap with the ground truth label images. We additionally compared the T-stage diagnostic accuracy by comparing the above two algorithm in this revision, and clearly stated the strength of this algorithm in the results section (page 19, line 311-page 20, line 315).

3.In line 148, proper citations are missing. Please place the appropriate citations for evaluation coefficients.

Response:

We decided that the indicated point was described in line 248. We used Dice similarity coefficients (DSC) to assess the segmentation accuracy, and in this revision, we cited a reference #25 to explain this method (page 17, line 265).

#25. Zou KH, Warfield SK, Bharatha A et al. Statistical validation of image segmentation quality based on a spatial overlap index. Acad Radiol 2004; 11: 178-189.

4. In figure 5, pearson correlation coefficient and p value should be noted in the figure,and the y-axis is fuzzy.

Response:

Thank you very much for your suggestions. Pearson correlation coefficient was 0.2418 and P value was 0.0081, which were added in the figure 5(c). In addition, y-axis “Dice similarity coefficients” was made clearly in this revision.

Additional revision

We cited a previous report from our institution showing the method for “transverse slicing of a semi-opened rectal specimen” in this revision (page 11, line 157).

#24. Ishii M, Takemasa I, Okita K et al. A modified method for resected specimen processing in rectal cancer: Semi-opened with transverse slicing for measuring of the circumferential resection margin. Asian J Endosc Surg 2021.

Reviewer #2:

 This manuscript presents a new high-resolution MRI semi-automatic rectal cancer segmentation technology based on deep learning. I believe this work will contribute to the accuracy of adjuvant therapy. However, I hope the author will revise the manuscript and solve the following problems before publishing it in the journal.

1. Firstly, the description of the research status is not perfect, especially the research status of deep learning in medical image aided diagnosis, including two-dimensional image and three-dimensional image.

Response:

We thank the reviewer for the constructive comment. We have added the current status regarding the tumor segmentation using 2D and 3D U-net in introduction section (page 7, line 99-105). We additionally cited the following 6 references herein.

#18. Ronneberger O, Fischer, P., Brox, T. U-Net: Convolutional Networks for Biomedical Image Segmentation. arXiv:1505.04597 2015.

#19. Milletari F, Navab, N., Ahmadi, S.A. V-Net: Fully Convolutional Neural Networks for Volumetric Medical Image Segmentation. arXiv:1606.04797.

#20. Dolz J, Xu X, Rony J et al. Multiregion segmentation of bladder cancer structures in MRI with progressive dilated convolutional networks. Med Phys 2018; 45: 5482-5493.

#21. Hodneland E, Dybvik JA, Wagner-Larsen KS et al. Automated segmentation of endometrial cancer on MR images using deep learning. Sci Rep 2021; 11: 179.

#22. Trebeschi S, van Griethuysen JJM, Lambregts DMJ et al. Author Correction: Deep Learning for Fully-Automated Localization and Segmentation of Rectal Cancer on Multiparametric MR. Sci Rep 2018; 8: 2589.

#23. Huang YJ, Dou Q, Wang ZX et al. 3-D RoI-Aware U-Net for Accurate and Efficient Colorectal Tumor Segmentation. IEEE Trans Cybern 2021; 51: 5397-5408.

2. Secondly, it is necessary to make a detailed description of the hardware configuration and super parameter configuration of network training.

Response: 

Thank you very much for your suggestions. The indicated comment is important, and we added the implementation details for network training as follows; The parameter λ in the loss function was experimentally determined to be 0.02. To minimize the loss function, the Adam optimizer with a base learning rate of 0.003, beta1 0.9, beta2 0.999, epsilon 1e-8. Batch-size was 5 samples which consisted of 3 cases with ground-truth segmentation labels and 2 cases with only ground-truth staging. All experiments are conducted on an NVIDIA DGX-2 machine using the NVIDIA V100 GPU with 80GBs of memory. During the training iteration, the performance of the network was evaluated every 100 iterations on the validation dataset. We chose the best network parameter for the validation dataset using the sum of dice score, sensitivity and specificity, then applied it to the evaluation dataset. This information was added in the materials and methods section (page 16, line 241, line 244-247, and page 17, line 259-260.).

3. Finally, some other networks should be added to the experimental part as a contrast, so as to make the experimental results more convincing.

Response:

According to the indication, we added the comparative analysis between general methods and ours. The salient feature of the algorithm is a novel loss which can directly maximize T-staging accuracies in model training. To reveal the effectiveness of the proposed loss function, we evaluated a baseline model which was trained by using a standard dice loss with the ground truth label images. We additionally compared the T-stage diagnostic accuracy by comparing the above two algorithms in this revision, and clearly stated the strength of this algorithm in the results section (page 14, line 213-218, and page 19, line 311-page 20, line 315).

Additional revision

We cited a previous report from our institution showing the method for “transverse slicing of a semi-opened rectal specimen” in this revision (page 11, line 157).

#24. Ishii M, Takemasa I, Okita K et al. A modified method for resected specimen processing in rectal cancer: Semi-opened with transverse slicing for measuring of the circumferential resection margin. Asian J Endosc Surg 2021.

---

## [Decision Letter · Decision Letter 1]

1 Jun 2022

Artificial intelligence–based technology for semi-automated segmentation of rectal cancer using high-resolution MRI

PONE-D-21-25683R1

Dear Dr. Takemasa,

We’re pleased to inform you that your manuscript has been judged scientifically suitable for publication and will be formally accepted for publication once it meets all outstanding technical requirements.

Kind regards,

Kumaradevan Punithakumar

Academic Editor

PLOS ONE

Additional Editor Comments (optional):

Reviewers' comments:

Reviewer's Responses to Questions

**Comments to the Author**

1. If the authors have adequately addressed your comments raised in a previous round of review and you feel that this manuscript is now acceptable for publication, you may indicate that here to bypass the “Comments to the Author” section, enter your conflict of interest statement in the “Confidential to Editor” section, and submit your "Accept" recommendation.

Reviewer #2: All comments have been addressed

2. Is the manuscript technically sound, and do the data support the conclusions?

Reviewer #2: Yes

3. Has the statistical analysis been performed appropriately and rigorously? 

Reviewer #2: Yes

4. Have the authors made all data underlying the findings in their manuscript fully available?

Reviewer #2: Yes

5. Is the manuscript presented in an intelligible fashion and written in standard English?

Reviewer #2: Yes

7. PLOS authors have the option to publish the peer review history of their article (what does this mean?). If published, this will include your full peer review and any attached files.

Reviewer #2: No

---

## [Editor Report · Acceptance letter]

9 Jun 2022

PONE-D-21-25683R1 

Artificial intelligence–based technology for semi-automated segmentation of rectal cancer using high-resolution MRI 

Dear Dr. Takemasa:

I'm pleased to inform you that your manuscript has been deemed suitable for publication in PLOS ONE. Congratulations! Your manuscript is now with our production department. 

Kind regards, 

on behalf of

Professor Kumaradevan Punithakumar 

Academic Editor

PLOS ONE